# Allosteric Inhibition of c-Abl to Induce Unfolded Protein Response and Cell Death in Multiple Myeloma

**DOI:** 10.3390/ijms232416162

**Published:** 2022-12-18

**Authors:** Hideki Kosako, Yusuke Yamashita, Shuhei Morita, Sadahiro Iwabuchi, Shinichi Hashimoto, Taka-Aki Matsuoka, Takashi Sonoki, Shinobu Tamura

**Affiliations:** 1Department of Hematology/Oncology, Wakayama Medical University, Wakayama 641-8509, Japan; 2The First Department of Internal Medicine, Wakayama Medical University, Wakayama 641-8509, Japan; 3Department of Molecular Pathophysiology, Wakayama Medical University, Wakayama 641-8509, Japan; 4Department of Emergency and Intensive Care Medicine, Wakayama Medical University, Wakayama 641-8509, Japan

**Keywords:** unfolded protein response, IRE1α, c-Abl, GNF-2, asciminib, multiple myeloma

## Abstract

Endoplasmic reticulum stress activates inositol-requiring enzyme 1α (IRE1α) and protein kinase, R-like endoplasmic reticulum kinase (PERK), the two principal regulators of the unfolded protein response (UPR). In multiple myeloma, adaptive IRE1α signaling is predominantly activated and regulates cell fate along with PERK. Recently, we demonstrated that GNF-2, an allosteric c-Abl inhibitor, rheostatically enhanced IRE1α activity and induced apoptosis through c-Abl conformational changes in pancreatic β cells. Herein, we analyzed whether the pharmacological modulation of c-Abl conformation resulted in anti-myeloma effects. First, we investigated the effects of GNF-2 on IRE1α activity and cell fate, followed by an investigation of the anti-myeloma effects of asciminib, a new allosteric c-Abl inhibitor. Finally, we performed RNA sequencing to characterize the signaling profiles of asciminib. We observed that both GNF-2 and asciminib decreased cell viability and induced *XBP1* mRNA splicing in primary human myeloma cells and myeloma cell lines. RNA sequencing identified the induction of UPR- and apoptosis-related genes by asciminib. Asciminib re-localized c-Abl to the endoplasmic reticulum, and its combination with a specific IRE1α inhibitor, KIRA8, enhanced cell death with the reciprocal induction of *CHOP* mRNA expression. Together, the allosteric inhibition of c-Abl-activated UPR with anti-myeloma effects; this could be a novel therapeutic target for multiple myeloma.

## 1. Introduction

Endoplasmic reticulum (ER) stress, caused by a static or excessive demand for protein folding, including the accumulation of proteins in the ER, induces the unfolded protein response (UPR) [1,2]. Under remediable ER stress, UPR reduces cellular stress and maintains homeostasis by activating signaling pathways initiated by stress sensors on the ER membrane, known as adaptive UPR (A-UPR). However, under irremediable ER stress, the dysregulation of cell function or cell fate is induced by the regulation of ER membrane residual key UPR sensors, such as inositol-requiring enzyme 1α (IRE1α) or protein kinase R-like endoplasmic reticulum kinase (PERK); this is known as terminal UPR (T-UPR) [3].

In several human cancers, A-UPR is predominantly activated to induce cell survival and maintenance under microenvironmental stresses such as hypoxia and low nutritional status [4,5,6,7]. In particular, in secretory cancers, such as multiple myeloma and neuroendocrine tumors, it is speculated that the cells may be susceptible to UPR due to the excessive production of secretory proteins and several cancer-related stimulations [6]. Indeed, we and other research groups have revealed the phenotypes of dominantly adaptive IRE1α signaling that induce A-UPR, which contributes to secretory cancer progression and tumor growth [8,9,10]. Furthermore, we and other research groups have focused on the loss of adaptive UPR and the imbalance of IRE1α and PERK signaling to induce cell death/apoptosis by the newly developed IRE1α specific inhibitor, KIRA8 [8,10]. Therefore, the regulation of the UPR has recently been highlighted as a fascinating novel target against secretory cancer cells, including multiple myeloma (MM) [6,7,10].

The *BCR-ABL* fusion oncogene product is produced in patients with certain types of leukemia. Tyrosine kinase inhibitors (TKIs) targeting BCR-ABL (c-Abl kinase) have been developed [11] and approved as the standard therapy for such patients [12]. However, mutations in *BCR-ABL* lead to resistance against the currently approved TKIs. Notably, chronic myeloid leukemia (CML) patients with the T315I mutation have been reported to have poorer prognoses compared to patients without the mutation [13,14,15]. Moreover, because these TKIs target not only c-Abl kinase but also other multiple kinase families, TKI-treated patients experience certain adverse events [16]. Therefore, a therapeutic agent with a novel mechanism against the BCR-ABL protein needs to be developed. GNF-2 was first discovered as a highly selective, allosteric, and non-ATP competitive inhibitor of TKI-resistant ATP-site mutations; however, this drug exhibits insufficient pharmacological effects [17]. Subsequently, asciminib, which has a structure similar to GNF-2, was developed. Asciminib induces a conformational change in the BCR-ABL protein and has been known as a Food and Drug Administration (FDA)-approved agent for adults with CML in the chronic phase (CP) [18,19,20].

In mammalian cells, c-Abl is a signaling molecule that predominantly localizes to the cytoplasm and regulates cell proliferation, differentiation, adhesion, and migration [21]. Under ER stress, we have recently shown that cytosol-anchored c-Abl migrates to the ER membrane and scaffolds to IRE1α, which rheostatically enhances the kinase/RNase activities of IRE1α and induces T-UPR in β cells [3]. Nilotinib and imatinib, selective c-Abl tyrosine kinase inhibitors, have been shown to inhibit IRE1α activation, suppress ER stress-induced β cell death, and reverse autoimmune diabetes in mice [3]. Meanwhile, GNF-2, an allosteric c-Abl inhibitor, has been shown to enforce c-Abl to migrate to the ER membrane and to scaffold IRE1α to induce T-UPR [3].

The introduction of novel drugs, such as proteasome inhibitors, immunomodulatory drugs, and monoclonal antibodies for the treatment of MM, has led to significant improvements in patient outcomes [22]. However, despite these therapeutic approaches, MM remains an incurable disease and refractory to these drugs, with a poor prognosis [23,24]. Accordingly, drug targets for novel signaling pathways in myeloma cells can provide further improvements in prognosis and are being actively developed [25,26,27]. Recently, we have demonstrated that nilotinib and KIRA8 exhibited anti-myeloma effects by targeting adaptive IRE1α signaling and modulating PERK signaling to induce T-UPR [10].

This motivated us to investigate the effect(s) of the pharmacological modulation of c-Abl on MM. This study aimed to clarify whether and how allosteric c-Abl inhibitors exhibited anti-myeloma effect(s) in the context of UPR signaling. The findings of the present study suggest c-Abl-UPR signaling as a novel therapeutic target in MM.

## 2. Results

### 2.1. Decreased Cell Viability and Induction of XBP1 Splicing by GNF-2 in Primary Myeloma Cells Derived from Patients with Multiple Myeloma

Figure 1a displays the structure of GNF-2, an allosteric c-Abl inhibitor that selectively binds to the myristate pocket of BCR-ABL [17]. The expression of c-Abl and IRE1α in primary myeloma cells from patients was verified by Western blotting (Appendix A). To determine the effects of GNF-2 on MM, we first used CD138-positive primary myeloma cells isolated from the bone marrow of three patients with newly diagnosed MM (NDMM) (Patients #1-3). Cell viability was determined 48 h after the treatment with GNF-2 or vehicle. The viability of myeloma cells decreased in a dose-dependent manner (Figure 1b). The mean value obtained from three independent experiments shows that 20 μM GNF-2 significantly decreased the cell viability (Figure 1c). Furthermore, as we have previously demonstrated the effect of GNF-2 on the activation of IRE1α in β-cells [3], the splicing of *XBP1* mRNA was induced by GNF-2 in primary myeloma cells (Figure 1d). Quantitative RT-PCR revealed that the expression of the spliced form of *XBP1* (*sXBP1*) mRNA was also increased by GNF-2 (Figure 1e), confirming the activation of IRE1α RNase. Together, GNF-2 exhibited antitumor effects along with the activation of IRE1α in primary myeloma cells derived from NDMM patients.

### 2.2. Anti-Tumor Effects of GNF-2 in Human Myeloma Cells

To further investigate the anti-myeloma effects of GNF-2, we used several human myeloma cell lines, namely KMS-12-PE, KMS-11, KMS-11/BTZ, and IM-9. Notably, the KMS-11/BTZ cell line is refractory to bortezomib, an effective agent for treating MM in clinical practice [28]. Western blotting was performed to detect c-Abl protein expression in these myeloma cell lines. c-Abl protein was expressed to roughly the same extent in all cell lines, while IRE1α protein expression varied among these cell lines (Figure 2a). Next, the cell viability was assessed using the Cell Counting Kit-8 (CCK-8) assay after treatment with the indicated concentrations of GNF-2 for 72 h. Three MM cell lines (KMS-12-PE, KMS-11/BTZ, and IM-9) showed a significant decrease in viability when treated with >5 μM GNF-2 (Figure 2b). KMS-11 also showed a significant decrease in cell viability after treatment with 10 μM GNF-2 (Figure 2b). We further evaluated the apoptotic effect using annexin V/propidium iodide (PI) staining. Treatment with 20 μM GNF-2 significantly induced apoptosis in all four cell lines (Figure 2c,d). Moreover, GNF-2 treatment induced *XBP1* mRNA splicing in KMS-12-PE cells in a time- and dose-dependent manner (Figure 2e,f), supporting the direct activation of IRE1α upon GNF-2 treatment. Recently, we reported that KIRA8, a selective inhibitor of the IRE1α kinase domain, inhibited *XBP1* mRNA splicing [3,10]. After pretreatment with KIRA8 for 2 h, GNF-2 did not induce *XBP1* mRNA splicing in KMS-12-PE cells (Figure 2g). These findings suggest that GNF-2 decreases cell viability and induces apoptosis in human myeloma cell lines via c-Abl-mediated IRE1α activation.

### 2.3. Asciminib Exhibits Anti-Myeloma Effects

Asciminib was discovered as a strong allosteric inhibitor of the proliferation of BCR-ABL-expressing leukemia cells with the same mechanism as GNF-2, which bind to the myristate binding site, and was approved in October 2021 by the FDA to treat adult patients with CML-CP [18,19,20]. Figure 3a shows the structure of asciminib [18,19]. After GNF-2, we investigated whether asciminib exerted anti-myeloma effects. Similar to GNF-2, asciminib decreased cell viability in a dose-dependent manner in primary myeloma cells derived from patients with NDMM (Patients #4-6) (Figure 3b,c) and induced the splicing of *XBP1* mRNA (Figure 3d). Notably, treatment with the indicated concentrations of asciminib for 72 h led to a decrease in the viability of the four myeloma cell lines (Figure 3e). In KMS-12-PE cells, treatment with asciminib at a concentration of more than 10 μM significantly induced apoptosis (Figure 3f). Asciminib also induced the time- and dose-dependent splicing of *XBP1* mRNA in KMS-12-PE cells (Figure 3g,h). Moreover, in the parallel experiment, a decrease in cell viability was similarly observed in KMS-12-PE cells treated with GNF-2 and asciminib (Appendix A). These findings suggest that asciminib exerts anti-myeloma effects and leads to IRE1α activation, similar to GNF-2.

### 2.4. Gene Expression Profile Induced by Asciminib in Myeloma Cells

To further elucidate the anti-myeloma effects of asciminib, we performed RNA sequencing analysis (RNA-Seq) for the asciminib-treated KMS-12-PE cell line. The samples were collected in triplicate from cells treated with DMSO (control) or 10 μM asciminib for 2 h. This analysis revealed a total of 57,773 differentially expressed genes (DEGs) in the asciminib-treated cells compared to the control cells (Appendix A). To find the potential functions of these DEGs, we identified gene signatures regulated by asciminib using an unbiased gene set enrichment analysis (GSEA). This analysis classified 41 gene sets as upregulated gene sets and nine gene sets as downregulated gene sets. Significantly, the NF-κB pathway, JAK-STAT3 pathway, p53 pathway, UPR, and hypoxia were classified as upregulated gene sets, and the G2M checkpoint was classified as a downregulated gene set (nominal *p*-value < 0.05) (Figure 4a). Subsequently, we focused on UPR (enrichment score = 1.34; nominal *p*-value  =  0.027) and apoptosis (enrichment score = 1.21; nominal *p*-value  =  0.10) in the upregulated gene sets and validated their RNA-Seq data (Figure 4b,c). While *XBP1* gene expression was slightly induced by asciminib, *ATF4*, *ATF3*, and *CHOP* (also known as *DDIT3*) were also found to be upregulated among the top ten UPR- and apoptosis-related genes (Figure 4b,c). Notably, CHOP is a part of the PERK-eIF2α signaling pathway and induces apoptosis in cancer cells [6,7]. These RNA-Seq data suggest that asciminib induces gene signatures associated with global UPR activation in myeloma cells.

### 2.5. Trafficking of c-Abl to ER and Induction of Global UPR by Asciminib

To further confirm asciminib-induced UPR in myeloma cells, we investigated the expression levels of transcription factors associated with ER stress in KMS-12-PE cells treated with asciminib. After treatment with 10 μM asciminib, *CHOP* mRNA expression significantly increased at 120 min (Figure 5a). Next, we examined the changes in representative ER stress markers, including *CHOP, ATF4, sXBP1,* and *ATF6*, using quantitative RT-PCR [6,7]. Four hours after the treatment with 10 μM asciminib, the gene expression of these markers was found to have been significantly increased (Figure 5b). Among these markers, *CHOP* mRNA expression was increased the most, i.e., by over fourfold, upon treatment with asciminib (Figure 5b).

Motivated by the global activation of UPR by asciminib and previous reports showing the GNF-2-mediated translocation of c-Abl to the ER and induced T-UPR, we hypothesized that asciminib could enforce c-Abl relocation to the ER regardless of its kinase activity [3,29,30]. To assess this hypothesis, we performed immunocytochemical analysis using HEK293T cells overexpressing full-length (WT) or kinase-dead (T735A) c-Abl. Under control conditions, cytoplasmic WT c-Abl was ubiquitously overexpressed in HEK293T cells (Figure 5c). After treatment with asciminib, WT c-Abl relocated to membrane structures, including the ER (Figure 5c). In the ER, WT c-Abl and calreticulin were co-localized and aggregated, supporting the activation of UPR sensor proteins. Similarly, kinase-dead T735A c-Abl was also relocated and aggregated in the ER upon treatment with asciminib (Figure 5c). These results suggest that asciminib re-localizes c-Abl to the ER independent of its kinase activity.

Finally, to further investigate the anti-apoptotic mechanism of asciminib underlying global UPR activation and to demonstrate its novel therapeutic potential, we first attempted to demonstrate the effects of a global ER stress inducer thapsigargin (Tg) combined with a selective IRE1α inhibitor, KIRA8. KIRA8 almost completely shut down *XBP1* mRNA splicing induced by Tg in KMS-12-PE cells (Figure 5d,e). Conversely, the *CHOP* gene expression was increased by KIRA8 under ER stress, suggesting a reciprocal induction of the PERK pathway (Figure 5f). Finally, the combination of Tg with KIRA8 decreased the viability of KMS-12-PE cells compared to that upon treatment with a single agent (Figure 5g). Similarly, treatment with asciminib and KIRA8 completely inhibited *XBP1* mRNA splicing induced by asciminib (Figure 5h,i), induced *CHOP* expression (Figure 5j), and, finally, induced more cell death than that upon treatment with either of the agents (Figure 5k). Together, asciminib-induced global UPR and its combination with KIRA8 exhibited a strong anti-myeloma effect via an imbalance in the UPR.

## 3. Discussion

This is the first study to demonstrate that allosteric c-Abl inhibitors exhibit anti-tumor effects with UPR profiling in MM. Historically, BCR-ABL TKIs have been investigated for their anti-myeloma effects by targeting multiple family kinases. c-Kit is functionally expressed in 30% of MM patients, and imatinib, whose primary target is c-KIT, inhibited the proliferation of myeloma cells in vitro [31,32]. The second-generation BCR-ABL TKI, dasatinib, has also been shown to have preclinical efficacy for inhibiting the c-Src kinase family in myeloma cells [33,34]. Despite these preclinical investigations, BCR-ABL TKIs have never entered clinical practice. Recently, based on the accumulated evidence of the strong induction of adaptive IRE1α in MM patients and our findings showing the inhibitory effect of another second-generation BCR-ABL TKI, nilotinib, on IRE1α in pancreatic β cells [3], we first demonstrated the anti-myeloma effects of nilotinib, which were possibly facilitated by inhibiting adaptive IRE1α and reciprocally inducing pro-apoptotic CHOP via the PERK pathway [10]. Furthermore, we demonstrated that the effect of nilotinib on IRE1α was due to the conformational changes in c-Abl, but not due to c-Abl kinase inhibition. This motivated us to investigate whether (hyper-)induction of IRE1α/UPR by modulating c-Abl conformation could exhibit anti-myeloma effects.

To test our hypothesis, we first selected GNF-2, an allosteric inhibitor of c-Abl, as a tool. In contrast to nilotinib, we previously demonstrated that GNF-2 modulates c-Abl conformation to activate IRE1α and induce T-UPR and cell death in pancreatic β cells [3]. This allosteric inhibitor does not stimulate multiple family kinases, including c-Abl, which limits the off-target activity [17]. We confirmed the anti-tumor effects of GNF-2 and the activation of IRE1α in human primary myeloma cells and human myeloma cell lines. Thus, using GNF-2, we demonstrated that the UPR activation of an allosteric c-Abl inhibitor could serve as a potential therapeutic target for MM.

Motivated by the promising effect of GNF-2 against MM, we further explored the anti-myeloma effects of asciminib, the first allosteric c-Abl inhibitor in humans [18,19,20]. Asciminib binds to the myristate binding site with a higher affinity than GNF-2 [19] and has recently been introduced in clinical practice [19,20]. As expected, asciminib activated IRE1α by significantly inducing *XBP1* mRNA splicing in a dose- and time-dependent manner, as observed for GNF-2. In contrast to the IRE1α-dominant UPR-induction effect of GNF-2 in pancreatic β cells, a comprehensive RNA-profiling analysis revealed the global activation of UPR by asciminib in myeloma cells. These data suggest that asciminib could exhibit anti-myeloma effects based on its actions on not only IRE1α signaling but also PERK apoptotic signaling. Furthermore, our data show the additional anti-tumor effect of asciminib when combined with KIRA8, which supports previous studies showing the critical role of the UPR balance with IRE1α and PERK signaling in maintaining secretory cancers, which are susceptible to ER stress [8,10]. Together, our findings demonstrate the anti-tumor effect of asciminib by facilitating the global induction of UPR and the imbalance of UPR with KIRA8 in adaptive UPR-dominant MM cells.

Under the homeostatic condition, it is known that the c-Abl protein preferentially localizes to cytosol in mammalian cells [3]. Under ER stress, c-Abl re-localizes onto IRE1α at the ER membrane, which allows c-Abl to rheostatically enhance IRE1α activities to induce T-UPR and apoptosis, as demonstrated in *Abl1/Abl2* double knockout mouse embryonic fibroblasts [3]. Similar to ER stress, we previously demonstrated that GNF-2, which changes the c-Abl conformation to expose the myristoylation site, allows cytosol-anchored c-Abl to forcibly re-localize to the ER and induce T-UPR and apoptosis without c-Abl kinase phosphorylation in pancreatic β cells [3,29,30]. Our current findings provide new evidence that the FDA-approved allosteric inhibitor asciminib, which also changes c-Abl conformation to expose the myristoylation site, re-localizes c-Abl to the ER, and induces global UPR regardless of its kinase activity in myeloma cell lines.

In conclusion, we have demonstrated the anti-myeloma effect of the allosteric inhibition of c-Abl by GNF2 and asciminib in primary human myeloma cells and myeloma cell lines. We have also provided evidence that the anti-myeloma effect of allosteric inhibitors is associated with the global activation of UPR and that this effect is further enhanced by the imbalance of UPR. Asciminib was recently approved for adult patients with CML-CP, and it can potentially prevent side effects owing to its low off-target activity [20]. Considering the clinical efficacy and tolerable safety profile, an allosteric c-Abl inhibitor can potentially be used as a part of current combination regimens, which include bortezomib and lenalidomide, against MM. The regulatory mechanisms of c-Abl must be explored further to provide a basis for the clinical application of related drugs in the treatment strategies against MM.

## 4. Materials and Methods

### 4.1. Primary Myeloma Cells

Primary myeloma cells were isolated from the bone marrow of NDMM patients at Wakayama Medical University (Wakayama, Japan) between February 2022 and October 2022. Primary myeloma cells were purified using anti-CD138-coated magnetic MicroBeads and MACS columns (Miltenyi Biotec, Auburn, CA, USA); these cells were identified using CD138-PE staining (PE anti-human CD138 antibody from BioLegend, San Diego, CA, USA; Cat# 356503, RRID: AB_2561877) according to the supplier’s protocol and analyzed using FACS Verse (BD Biosciences, San Jose, CA, USA). These cells were cultured using RPMI-1640 (Sigma-Aldrich, St. Louis, MO, USA) supplemented with 10% fetal bovine serum (FBS) (Gibco, Grand Island, NY, USA), 100 U/mL penicillin, and 100 μg/mL streptomycin (Gibco, Grand Island, NY, USA) in an atmosphere containing 5% CO_2_ at 37 °C. Sample collection was approved by the hospital’s ethics committee (approval number: 2910, approved on 28 September 2021), and written informed consent was obtained from all study participants. This study was conducted in accordance with the Declaration of Helsinki guidelines. Appendix A summarizes the clinical characteristics of NDMM patients.

### 4.2. Cell Lines and Cell Culture

The human myeloma cell lines, IM-9 (RRID: CVCL_1305) and KMS-11/BTZ (RRID: CVCL_4V71), were purchased from the Japanese Collection of Research Bioresources Cell Bank/National Institute of Health Sciences (Tokyo, Japan). KMS-11 (RRID: CVCL_2989) and KMS-12-PE (RRID: CVCL_1333) cells were kindly provided by Dr. Hata and Dr. Kawano, Department of Hematology, Kumamoto University (Kumamoto, Japan) [35]. The cell lines were cultured as described previously [10]. The human kidney epithelial cell line HEK293T (RRID: CVCL_0063) was kindly provided by Dr. Kaisho and Dr. Sasaki, Department of Immunology, Institute of Advanced Medicine, Wakayama Medical University (Wakayama, Japan). The HEK293T cells were cultured in DMEM (code 08459-64; Nacalai Tesque, Kyoto, Japan) containing 10% fetal bovine serum (26140-079; Gibco, Grand Island, NY, USA) at 37 °C with 5% CO_2_ and passaged by trypsinization when they reached >80% confluence. The culture medium was renewed every 2–3 days.

### 4.3. Reagents

KIRA8 (CAS#1630086-20-2), GNF-2 (CAS#778270-11-4), and thapsigargin (CAS#67526-95-8) were purchased from Sigma-Aldrich (St. Louis, MO, USA). Asciminib (CAS#1492952-76-7) was purchased from Selleck Chemicals (Houston, TX, USA). All reagents were dissolved in dimethyl sulfoxide (DMSO) to a final concentration of 10 mM as a stock solution and stored at −30 °C.

### 4.4. Cell Viability Assay

The viability of human myeloma cell lines was measured using the Cell Counting Kit-8 (CCK-8) from Dojindo (Tokyo, Japan). These cells were seeded in a 96-well plate at a density of 2 × 10^4^ cells per well. The cells were then treated, as indicated, with each reagent. CCK-8 solution (10 μL) was added to each well, and the plates were incubated for an additional 2 h at 37 °C. Optical densities at 450 and 650 nm were measured using a Corona plate reader SH-9000 (Hitachi, Tokyo, Japan). The viability of primary myeloma cells was assessed using the CellTiter-Glo^®^ 2.0 Cell Viability Assay Kit (Promega Corporation, Madison, WI, USA) according to the supplier’s protocol and analyzed using a Corona plate reader SH-9000 (Hitachi, Tokyo, Japan).

### 4.5. Apoptosis Assay by Flow Cytometry

Apoptosis was evaluated as described previously [10]. We determined the apoptosis of myeloma cells using the annexin V–FITC Apoptosis Detection Kit (APOAF; Sigma-Aldrich, St. Louis, MO, USA) according to the manufacturer’s instructions. The percentage of apoptotic cells was determined by flow cytometry (FACS Verse Flow Cytometer; BD Biosciences, San Jose, CA, USA). Data analysis was performed using FlowJo software v10.8.0 (Tree Star, Ashland, OR, USA). We defined annexin V-positive cells as those undergoing apoptosis.

### 4.6. Western Blot Analysis

Briefly, human myeloma cell lines were lysed using RIPA lysis buffer (Nacalai Tesque, Kyoto, Japan) and cOmplete™ Mini Protease Inhibitor Cocktail (Roche Diagnostics, Basel, Switzerland). The protein concentration was determined using the BCA Protein Assay Kit (Thermo Fisher Scientific, Tokyo, Japan). The protein extracts were mixed with Laemmli sample buffer containing 2-mercaptoethanol (196-11022; Wako, Osaka, Japan) and heated in a heat block at 100 °C for 5 min. The samples were separated by sodium dodecyl sulfate-polyacrylamide gel electrophoresis (SDS–PAGE) and transferred onto polyvinylidene fluoride (PVDF) membranes (RPN1416F; GE Healthcare, Chicago, IL, USA). The membranes were washed and blocked with DIG Wash and Block Buffer Set (Roche Diagnostics, Basel, Switzerland), according to the supplier’s protocol. The membranes were then incubated overnight at 4 °C with primary antibodies. The following primary antibodies were used: mouse anti-human c-Abl antibody (Santa Cruz Biotechnology, Dallas, TX, USA; Cat# sc-56887, RRID: AB_781732, 1:1000 dilution), rabbit anti-human IRE1α antibody (Cell Signaling Technology, Danvers, MA, USA; Cat# 3294, RRID: AB_823545, 1:1000 dilution), and mouse anti-human GAPDH antibody (Proteintech, Rosemont, IL, USA; Cat# 60004-1-Ig, RRID: AB_2107436, 1:5000 dilution). The following day, after three washes, the membranes were incubated with horseradish peroxidase (HRP)-conjugated goat anti-rabbit IgG polyclonal antibody (Bio-Rad, Hercules, CA, USA; Cat# 170-6515, RRID: AB_11125142, 1:1000 dilution) or HRP-conjugated rabbit anti-mouse IgG antibody (SouthernBiotech, Birmingham, AL, USA; Cat# 1080-05, RRID: AB_2734756, 1:1000 dilution) as secondary antibody at room temperature (RT) for 1 h. Protein bands were visualized using SuperSignal^®^ West Dura Substrate (Thermo Fisher Scientific, Tokyo, Japan) and detected using a LuminoGraph I Chemiluminescent Imaging System (ATTO, Tokyo, Japan). GAPDH was used as an internal control to ensure that equal amounts of proteins were loaded.

### 4.7. Extraction of Total RNA, Reverse Transcription Polymerase Chain Reaction, and Detection of XBP1 Splicing

RNA extraction and reverse transcription were performed as previously described [10]. Quantitative RT-PCR was performed, as described previously, using the primers listed in Appendix A [10]. *XBP1* mRNA processing was measured by amplifying *XBP1* cDNA with the following primers: *XBP1* 5′-AAACAGAGTAGCAGCTCAGACTGC-3′ and 5′-GGATCTCTAAAACTAGAGGCTTGGTG-3′. The PCR products of *XBP1* cDNA were digested with PstI, resolved on a 3% agarose gel, stained with ethidium bromide, and quantified by densitometry using ImageJ software version 1.51 k (National Institutes of Health, Bethesda, MD, USA) [3,10].

### 4.8. RNA Sequencing and Gene Expression Analysis

For RNA sequencing analysis, the total RNA was isolated as previously described [10]. Six cDNA libraries were constructed and sequenced by the Department of Molecular Pathophysiology at Wakayama Medical University (Wakayama, Japan). The RNA quality was evaluated using an Agilent 4200 TapeStation (Agilent Technologies, Santa Clara, CA, USA), and the RNA concentration was measured using a Qubit Fluorometer (Thermo Fisher Scientific, Tokyo, Japan). A total of 1846–3017 ng RNA from each culture condition was used, and libraries for sequencing were constructed using TruSeq Stranded mRNA (Illumina, San Diego, CA, USA) according to the manufacturer’s protocol. The number of libraries was estimated using the KAPA Library Quantification Kit (Roche Diagnostics, Basel, Switzerland). The average library size was 376–517 bp. High-throughput sequencing of the samples was performed using a NextSeq 500/550 High Output Kit v2.5 (Illumina, San Diego, CA, USA, 75 cycles pair-end, 40/40 cycles). The average number of sequence reads per sample was 19,374,043. The bulk-RNA sequencing results were analyzed using the CLC Genomics Workbench Version 12.0.2 (Filgen, Aichi, Japan). Gene expression abundance was represented using the reads per kilobase per million (RPKM) value. Gene expression analysis, including gene set enrichment analysis (GSEA), was conducted using GSEA v4.2.3 software and the Molecular Signature Database (MSigDB) v7.5.1. GSEA is a computational method that determines whether a priori-defined set of genes shows statistically significant, concordant differences between two biological states [36,37].

### 4.9. Transfection and Immunocytochemical Analysis

Transfection of the vectors and immunocytochemical analysis were performed using the following procedures. HEK293T cells were plated on chamber slides and cultured; these cells were transiently transfected with pcDNA5/FRT/TO: human c-Abl 1b vectors (WT, T735A, and empty), which were constructed in a previously reported study [3], using lipofectamine 2000 reagent (Invitrogen, Waltham, MA, USA). The medium was replaced with fresh medium containing 10 μM asciminib or DMSO 18 h after the transfection. Two hours after the medium change, the cells were fixed in ice-cold 100% methanol (RT, 5 min), followed by 4% paraformaldehyde (RT, 15 min). Immunostaining was performed as previously described [38]. The cells were permeabilized and blocked in 0.1 M PBS containing 0.3% Triton X-100 and 10% normal goat serum (RT, 1 h) and then incubated with primary antibody solution containing 1% bovine serum albumin and 0.3% Triton X-100 (4 °C, overnight). Primary antibodies were used at the following dilutions: mouse anti-human c-Abl antibodies (Santa Cruz Biotechnology, Dallas, TX, USA; Cat# sc-56887, RRID: AB_781732, 1:100 dilution); Alexa Fluor^®^ 488-conjugated rabbit anti-human calreticulin antibodies (Abcam, Toronto, ON, Canada; Cat# ab196158, RRID: AB_2069806, 1:500 dilution). The next day, the cells were washed in 0.1 M PBS and incubated in a secondary antibody solution containing 1% bovine serum albumin and 0.3% Triton X-100 (RT, 1 h). The following secondary antibody was used: Cy3-conjugated goat anti-mouse IgG polyclonal antibody (Jackson ImmunoResearch Labs, West Grove, PA, USA; Cat# 115-165-003, RRID: AB_2338680, 1:500 dilution). After washing with 0.1M PBS, the cells were incubated in 1:500 DAPI (D9542; Sigma-Aldrich, St. Louis, MO, USA) containing 0.3% Triton X-100 (RT, 5 min). After washing with 0.1M PBS, the cells were enclosed with the 90% glycerol mounting medium. Immunofluorescence images were acquired using a confocal microscope LSM900 (Carl Zeiss Meditec AG, Jena, Germany).

### 4.10. Statistical Analysis

All the values are expressed as the mean ± standard error of the mean (SEM). All the statistical analyses were performed using GraphPad Prism version 9.3.0 (GraphPad Software, San Diego, CA, USA) and JMP^®^ Pro 14 (SAS Institute Inc., Cary, NC, USA). In addition, Student’s *t*-test or one-way analysis of variance, followed by post-hoc Tukey’s test, was used to assess the statistical difference between two groups or between more than two groups, respectively, unless otherwise noted. Statistical significance was set at *p* < 0.05. Asterisks (*) indicate that the values are significantly different from those of the control (* *p* < 0.05, ** *p* < 0.01, *** *p* < 0.001). All experiments in this study were independently repeated at least three times. Sample numbers are indicated in the figures and figure legends. For further information regarding the statistical analysis, see the respective results section.

## Figures and Tables

**Figure 1 ijms-23-16162-f001:**
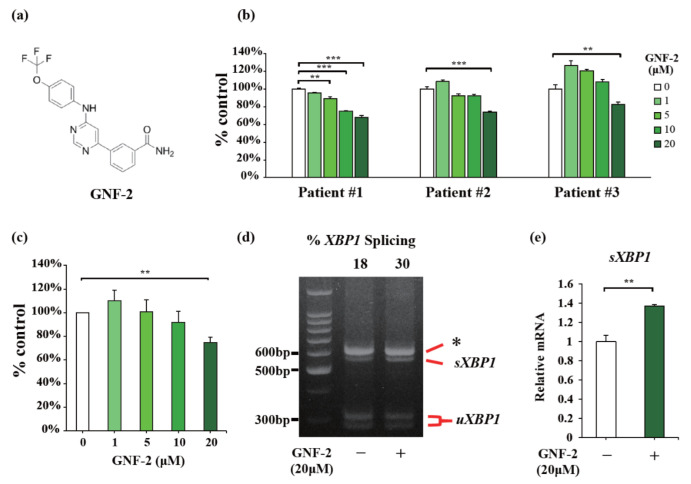
GNF-2 dose-dependently exerts anti-tumor effects in primary myeloma cells. (**a**) Structure of GNF-2. (**b**) CD138-positive cells from three patients with newly diagnosed multiple myeloma (NDMM) were cultured in the presence of GNF-2 (0–20 μM) for 48 h. In each case, the cell viability of triplicate cultures was assessed using the CellTiter-Glo^®^ 2.0 Cell Viability Assay Kit and expressed as the percentage of the value obtained for the untreated control. Data are presented as the mean ± SEM. (**c**) Mean viability value for CD138 positive cells obtained from three patients and cultured in the presence of GNF-2 (0–20 μM). Data are presented as the mean ± SEM. (**d**) PstI-digested *XBP1* cDNA amplicons from CD138 positive cells from one NDMM patient (#3) treated with 20 μM GNF-2 for 2 h. Quantified percentage of spliced *XBP1* (*sXBP1*) (**top**). The asterisk denotes the unspecific amplification. *uXBP1*; unspliced *XBP1*. (**e**) qPCR of relative *sXBP1* mRNAs from CD138 positive cells treated with GNF-2 (0 or 20 μM) for 2 h. Data were analyzed with one-way analysis of variance, followed by post-hoc Tukey’s test; * *p* < 0.05, ** *p* < 0.01, and *** *p* < 0.001.

**Figure 2 ijms-23-16162-f002:**
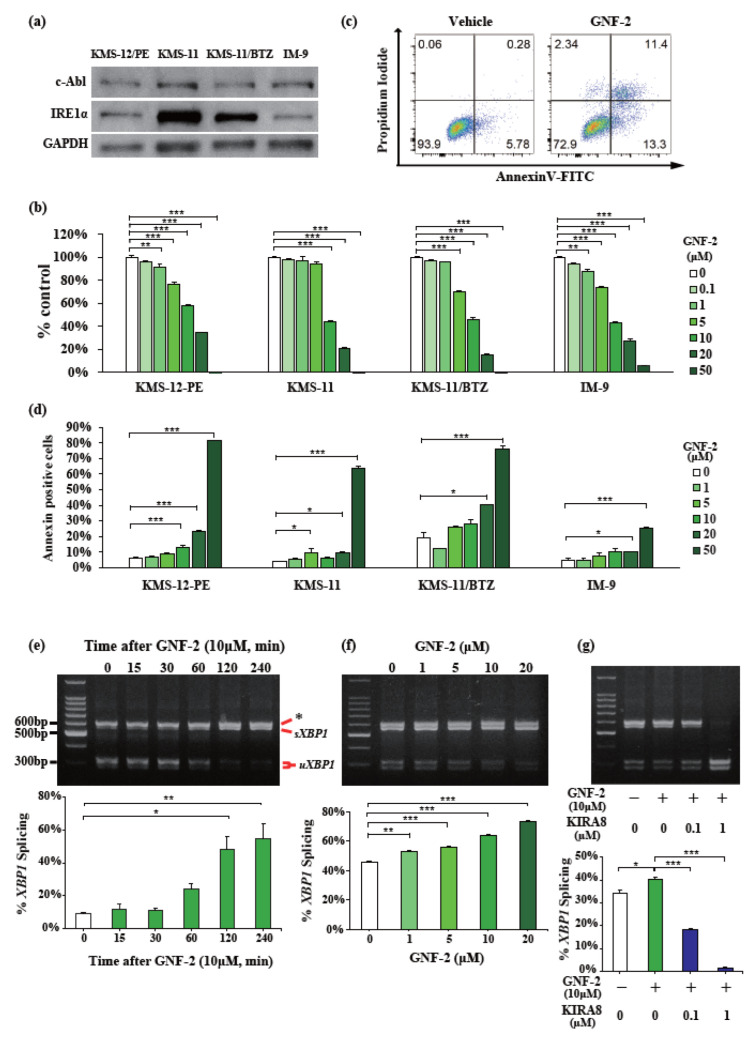
GNF-2 induces *XBP1* mRNA splicing and apoptosis in myeloma cell lines. (**a**) Whole-cell lysates of KMS-12-PE, KMS-11, KMS-11/BTZ, and IM-9 MM cell lines were subjected to Western blotting using antibodies against c-Abl, IRE1α, and GAPDH. (**b**) KMS-12-PE, KMS-11, KMS-11/BTZ, and IM-9 MM cell lines were treated with GNF-2 (0–50 μM) for 72 h. In each case, the cell viability of triplicate cultures was assessed using the Cell Counting Kit-8 (CCK-8) assay and is expressed as the percentage of the value obtained for the untreated control. (**c**,**d**) KMS-12-PE, KMS-11, KMS-11/BTZ, and IM-9 MM cell lines were cultured in the presence of GNF-2 (0–50 μM) for 24 h. In each case, apoptotic cells were analyzed with flow cytometry using annexin V/PI staining. Apoptosis was assessed as the percentage of annexin V-positive cells. Representative FACS plots of annexin V/PI-stained KMS-12-PE cells (**c**) and the percentage of apoptotic cells for each MM cell line (**d**). (**e**,**f**) PstI-digested *XBP1* cDNA amplicons from KMS-12-PE cells treated with 10 μM GNF-2 for indicated times (**e**) or indicated [GNF-2] for 2 h (**f**). Quantified percentage of spliced *XBP1* (**bottom**). (**g**) PstI-digested *XBP1* cDNA amplicons from KMS-12-PE cells were treated with indicated [KIRA8] for 2 h, followed by 10 μM GNF-2 for 2 h. Quantified percentage of spliced *XBP1* (bottom). Data were analyzed using one-way analysis of variance, followed by post-hoc Tukey’s test; * *p* < 0.05, ** *p* < 0.01, and *** *p* < 0.001.

**Figure 3 ijms-23-16162-f003:**
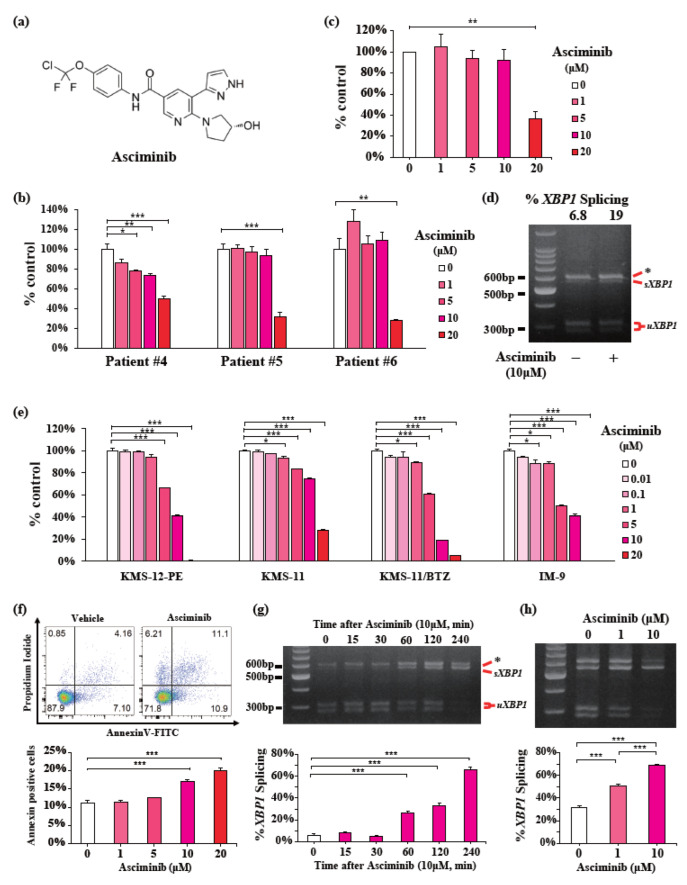
Asciminib dose-dependently exerts anti-myeloma effects in primary myeloma cells. (**a**) Structure of asciminib. (**b**) CD138-positive cells from three NDMM patients were cultured in the presence of asciminib (0–20 μM) for 48 h. The cell viability of triplicate cultures was assessed using the Cell Titer-Glo^®^ 2.0 Cell Viability Assay Kit and expressed as the percentage of the value obtained for the untreated control. Data are presented as the mean ± SEM. (**c**) Mean viability value for CD138-positive cells from three patients cultured in the presence of asciminib (0–20 μM). Data are presented as the mean ± SEM. (**d**) PstI-digested *XBP1* cDNAs amplicons from CD138-positive cells from one NDMM patient (#6) treated with 10 μM asciminib for 4 h. Quantified percentage of spliced *XBP1* (**top**). (**e**) KMS-12-PE, KMS-11, KMS-11/BTZ, and IM-9 MM cell lines were treated with asciminib (0–20 μM) for 72 h. In each case, the cell viability of triplicate cultures was assessed using the CCK-8 assay and expressed as the percentage of the value obtained for the untreated control. (**f**) KMS-12-PE MM cells were cultured in the presence of asciminib (0–20 μM) for 24 h. Apoptotic cells were analyzed with flow cytometry using annexin V/PI staining. Apoptosis was assessed as the percentage of annexin V-positive cells. Representative FACS plots of annexin V/PI-stained cells (**left**) and the percentage of apoptotic cells (**right**). (**g**,**h**) PstI-digested *XBP1* cDNA amplicons from KMS-12-PE cells treated with 10 μM asciminib for indicated times (**g**) or indicated [asciminib] for 4 h (**h**). Quantified percentage of spliced *XBP1* (**bottom**). Data were analyzed using one-way analysis of variance, followed by post-hoc Tukey’s test; * *p* < 0.05, ** *p* < 0.01, and *** *p* < 0.001.

**Figure 4 ijms-23-16162-f004:**
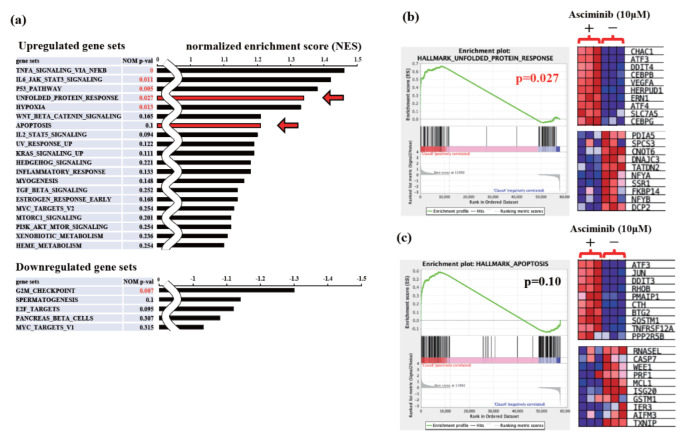
Asciminib triggers UPR in MM cells. KMS-12-PE cells were cultured with the vehicle (DMSO) or 10 μM asciminib for 2 h. Subsequently, RNA was extracted, and gene expression was analyzed. (**a**) Enrichment scores of excerpted top 20 upregulated (**upper**) and top 5 downregulated (**bottom**) enriched pathways using hallmark gene sets in the asciminib group compared with the vehicle group. UPR- and apoptosis-related gene sets, indicated by the red arrow, represent the focus of this study. (**b**,**c**) Gene set enrichment analysis (GSEA) using gene ontology (GO) datasets focusing on UPR- (**b**) and apoptosis-related (**c**) gene sets. “*p*” indicates nominal *p*-value. The heatmap shows the changes in gene expression in the gene lists of the top ten upregulated and downregulated genes within the UPR and apoptosis gene sets. NES, normalized enrichment score.

**Figure 5 ijms-23-16162-f005:**
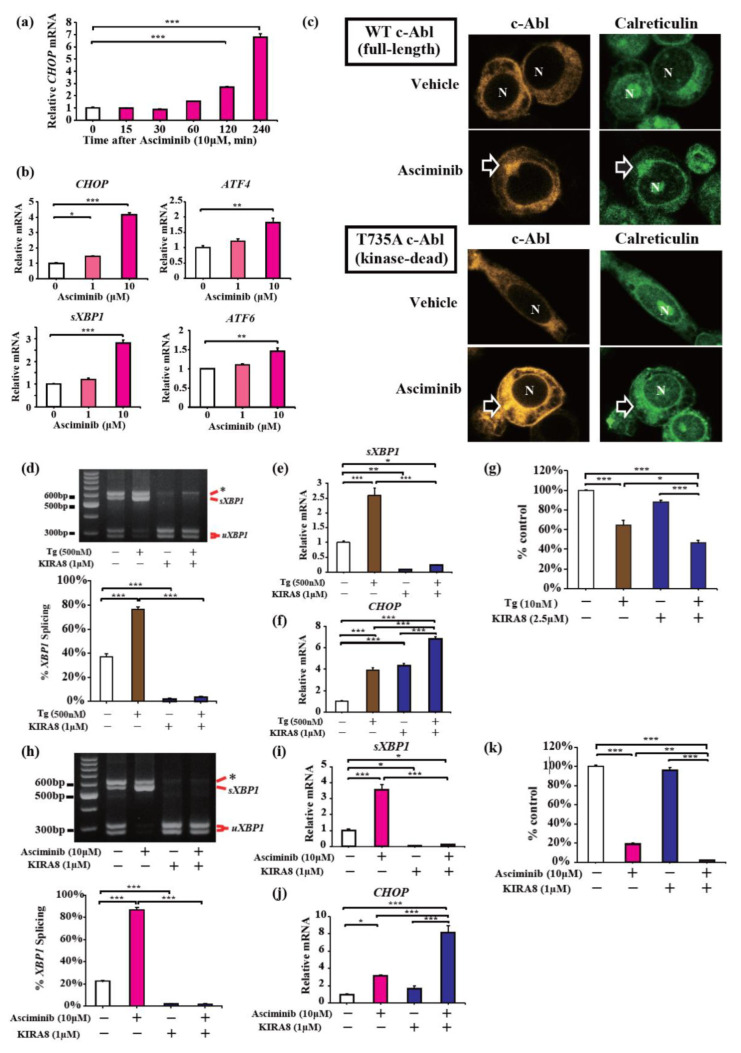
Asciminib induces the translocation of c-Abl to the ER and causes terminal UPR in MM cells. (**a**) qPCR analysis for relative *CHOP* mRNA expressions from KMS-12-PE cells treated with asciminib for indicated times. (**b**) qPCR analysis for assessing the relative mRNA expressions of UPR-related genes (*CHOP, ATF4, sXBP1,* and *ATF6*) from KMS-12-PE cells treated with indicated [asciminib] for 4 h. (**c**) Immunocytochemical analysis of HEK293T cells overexpressing c-Abl; N: Nucleus. The arrows indicate the co-localization and aggregation of c-Abl and calreticulin in the ER induced by exposure to asciminib. N: nucleus. (**d**) PstI-digested *XBP1* cDNA amplicons from KMS-12-PE cells treated for 4 h with thapsigargin (0 or 500 nM) and KIRA8 (0 or 1 μM). Quantified percentage of spliced *XBP1* (**bottom**). (**e**) qPCR analysis for relative *sXBP1* and (**f**) *CHOP* mRNAs from KMS-12-PE cells treated for 4 h with thapsigargin (0 or 500 nM) and KIRA8 (0 or 1 μM). (**g**) After KMS-12-PE cells were treated for 24 h with thapsigargin (0 or 10 nM) and KIRA8 (0 or 2.5 μM), the cell viability of triplicate cultures was assessed using the CCK-8 assay and is expressed as the percentage of the value obtained for the untreated control. (**h**) PstI-digested *XBP1* cDNA amplicons from KMS-12-PE cells were treated for 4 h with asciminib (0 or 10 μM) and KIRA8 (0 or 1 μM). Quantified percentage of spliced *XBP1* (**bottom**). (**i**) qPCR for assessing the relative expressions of *sXBP1* and (**j**) *CHOP* mRNAs from KMS-12-PE cells treated for 4 h with asciminib (0 or 10 μM) and KIRA8 (0 or 1 μM). (**k**) After KMS-12-PE cells were treated for 24 h with asciminib (0 or 10 μM) and KIRA8 (0 or 1 μM), the cell viability of triplicate cultures was assessed using the CCK-8 assay and is expressed as the percentage of the value obtained for the untreated control. Data were analyzed using one-way analysis of variance, followed by post-hoc Tukey’s test; * *p* < 0.05, ** *p* < 0.01, and *** *p* < 0.001.

## Data Availability

Not applicable.

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
