# Peer review of "Allosteric Inhibition of c-Abl to Induce Unfolded Protein Response and Cell Death in Multiple Myeloma"

_ijms, 2022, doi:10.3390/ijms232416162_

Round 1

Reviewer 1 Report

Dear Editor,

Dear Authors,

Thank you for inviting me to review the manuscript entitled Allosteric inhibition of c-Abl to induce unfolded protein 2 response and cell death in multiple myeloma by Hideki Kosako et al., that covers an interesting topic but, in my opinion, requires a major revision:

1. Keywords should facilitate the association with MeSH terms. You might consider using keywords that are not already used in the title of the article.

2. Sections “Introduction” and “Discussion” should be written based on the very recently papers published on that subject. Please revise the cited papers and include current papers from the last 4 years.

3. The literature references should be systematically revised. Please always cite the article that supports your claim with data and results. Random checks seem to indicate that this principle has not been respected. However, the bibliographical reference that is cited as the source does not contain the underlying facts at all, but only a similar wording and a reference to another work. Only those papers should be cited that support your own assertions based on results and conclusions. Introductions or reviews that merely refer to other work should not be cited here as the source, but the work in which the statement was actually supported by results. Example:

“The BCR-ABL fusion oncogene product is produced in patients with certain types of 60 leukemia. Tyrosine kinase inhibitors (TKIs) targeting BCR-ABL (c-Abl kinase) have been 61 developed and approved as the standard therapy for such patients”.

Authors described in this sentence the original data, whereas the source is listed as review paper: Rosti, G.; Castagnetti, F.; Gugliotta, G.; Baccarani, M. Tyrosine kinase inhibitors in chronic myeloid leukaemia: which, 555 when, for whom? Nat Rev Clin Oncol 2017, 14, 141-154. https://doi.org/10.1038/nrclinonc.2016.139.

4. Section “Materials and Methods” should be described in more details.

Author Response

To Reviewer 1:

Reviewer 1

Dear Authors,

Thank you for inviting me to review the manuscript entitled Allosteric inhibition of c-Abl to induce unfolded protein 2 response and cell death in multiple myeloma by Hideki Kosako et al., that covers an interesting topic but, in my opinion, requires a major revision:

> Thank you for your detailed reading of our manuscript. We are pleased that you found the study to cover an interesting topic. In the revision, we corrected the points that you suggested. The manuscript has significantly improved by incorporating your suggestions. With these changes, we hope that the manuscript is now acceptable for publication.

  1. Keywords should facilitate the association with MeSH terms. You might consider using keywords that are not already used in the title of the article.

> Response: As you suggested, two keywords, “inositol-requiring enzyme 1α” and “cytosolic Abl”, were difficult to find in the database. Thus, we changed “inositol-requiring enzyme 1α” to “IRE1α” and “cytosolic Abl” to “c-Abl”, respectively.

  1. Sections “Introduction” and “Discussion” should be written based on the very recently papers published on that subject. Please revise the cited papers and include current papers from the last 4 years.

> Response: As you pointed out, there were some references that were not recent. We have replaced them in the references section as follows.

“[4] Nat Rev Cancer 2014, 14, 581-597.” to “[4] Int J Mol Sci. 2019;20(3):749.” (Line 49, Page 2)

“[4] Nat Rev Cancer 2014, 14, 581-597.” and “[5] Cell 2017, 168, 692-706.” to “[10] Int. J. Mol. Sci. 2020, 21(17), 6314” (Line 58, Page 2)

“[13] Nat Rev Clin Oncol 2017, 14, 141-154.” to “[12] Am J Hematol. 2022 Sep;97(9):1236-1256.” (Line 61, Page 2)

“[22] J Clin Oncol. 2019 May 10;37(14):1228-1263.” added. (Line 85, Page 2)

“[24] Leukemia. 2022 Mar;36(3):873-876.” Added. (Line 87, Page 2)

  1. The literature references should be systematically revised. Please always cite the article that supports your claim with data and results. Random checks seem to indicate that this principle has not been respected. However, the bibliographical reference that is cited as the source does not contain the underlying facts at all, but only a similar wording and a reference to another work. Only those papers should be cited that support your own assertions based on results and conclusions. Introductions or reviews that merely refer to other work should not be cited here as the source, but the work in which the statement was actually supported by results. Example:

“The BCR-ABL fusion oncogene product is produced in patients with certain types of 60 leukemia. Tyrosine kinase inhibitors (TKIs) targeting BCR-ABL (c-Abl kinase) have been 61 developed and approved as the standard therapy for such patients”.

Authors described in this sentence the original data, whereas the source is listed as review paper: Rosti, G.; Castagnetti, F.; Gugliotta, G.; Baccarani, M. Tyrosine kinase inhibitors in chronic myeloid leukaemia: which, 555 when, for whom? Nat Rev Clin Oncol 2017, 14, 141-154. https://doi.org/10.1038/nrclinonc.2016.139.

> Response: Thank you for your comments. As suggested, we have carefully re-evaluated all the cited articles and replaced the inappropriate references to the suitable ones. Here is the list of the revised references:

“[8] Blood 2010, 116, 250-253.” and “[9] Haematologica 2014, 99, e14-16.” deleted. (Line 54, Page 2)

“[11] J Clin Invest. 2000 Jan;105(1):3-7.” added. (Line 61, Page 2)

“[13-15]” to “[16] Int J Hematol. 2021 May;113(5):632-641.” (Line 66, Page 2)

“[16] Nat Chem Biol 2006, 2, 95-102.” Deleted. (Line 73, Page 2 and Line 169, Page 5)

“[4] Nat Rev Cancer 2014, 14, 581-597.” and “[5] Cell 2017, 168, 692-706.” Deleted. (Line 199, Page 6 and Line 239, Page 7)

“[32] Leuk Res 2020, 98, 106458.” to “[17] Nat Chem Biol 2006, 2, 95-102.” (Line 290, Page 8)

“[16] Nat Chem Biol 2006, 2, 95-102.” and “[17] J Med Chem 2018, 61, 8120-8135.” Deleted. (Line 297, Page 9)

“[19, 21-24]” deleted. (Line 350, Page 10)

Reviewer 2 Report

The paper by Kosako and colleagues reports a study focused on the evaluation of the effects c-Abl inhibitors on multiple myeloma primary cells and cell lines.

The study presents several major concerns that need to be addressed before publication:

-The part relative to GNF-2 testing is not well integrated with the rest of the manuscript: does GNF-2 share the same mechanism of action of ascinimib? Why GNF-2 has not been tested in the same primary cells or cell lines as asciminib? Parallel experiment should be performed for both compounds, or otherwise, this part should be removed from the manuscript.

-In overall, the introduction is not clear and quite misleading: is activation or inactivation of IRE1a  required for apoptosis? Authors should clearly specify that the loss adaptive UPR leads to apoptosis (ref 10 of the present manuscript).

-Figure 4: Up-regulation of apoptosis-related gene set is not statistically significant

-Expression of c-Abl and IRE1a should be assessed in MM primary cells

-Line 267. The term unique is not appropriate since no other c-Abl inhibitors have been compared in this study

Author Response

To Reviewer 2:

Reviewer 2

The paper by Kosako and colleagues reports a study focused on the evaluation of the effects c-Abl inhibitors on multiple myeloma primary cells and cell lines.

The study presents several major concerns that need to be addressed before publication:

> Thank you for your detailed reading and suggestions on our manuscript. We are pleased that you appreciated the major instructive results of the work and recommended the publication of a revised version. Your suggestions and comments were very helpful, and during the revision phase, we made every attempt to address your questions by further analysis of the data or through clarifications in the narrative (please see point-by-point responses, below). Through addressing your suggestions, the manuscript is now much improved. With these changes, we hope that the manuscript is now acceptable to you for publication.

As you suggested, this manuscript has been proofread by an MDPI English Editing Services in this revision phase.

-The part relative to GNF-2 testing is not well integrated with the rest of the manuscript: does GNF-2 share the same mechanism of action of ascinimib? Why GNF-2 has not been tested in the same primary cells or cell lines as asciminib? Parallel experiment should be performed for both compounds, or otherwise, this part should be removed from the manuscript.

> Response: Thank you for highlighting the important point. To clarify the same mechanism by which two small molecules act, we have added the sentence “with the same mechanism as GNF-2” in the result section. Additionally, we conducted a parallel experiment of both compounds. In this parallel experiment, a decrease in cell viability was similarly observed in KMS-12-PE cells with the treatment of each compound. We have added these additional results as supplemental data (Figure S2) and modified the related parts for the readers to easily understand in the main text (Line 166-169, Page 2).

-In overall, the introduction is not clear and quite misleading: is activation or inactivation of IRE1a required for apoptosis? Authors should clearly specify that the loss adaptive UPR leads to apoptosis (ref 10 of the present manuscript).

> Response: In response to your comments, we and other groups have demonstrated the following concepts: In secretory cancers, the global ER stress, including the activation of adaptive IRE1α, is observed. Under these global ER stress, IRE1α specific inhibitor KIRA8 could imbalance the UPR activity to induce reciprocal activation of PERK, which leads to the terminal UPR from the PERK arm side like CHOP induction. Meanwhile, as you pointed out, the depletion of adaptive UPR can also contribute to inducing apoptosis. To clarify these two points, we have modified the related part in the second paragraph in the introduction section of the main text (Line 54-56, Page 2).

-Figure 4: Up-regulation of apoptosis-related gene set is not statistically significant

> Response: As you pointed out, the apoptosis-related gene set was upregulated, but not statistically significant.

We have deleted the phrase “significant” in the related part of the abstract (Line 30, Page 1).

-Expression of c-Abl and IRE1a should be assessed in MM primary cells

> Response: In response to your comment, we performed an additional experiment on the expression of c-Abl and IRE1α in primary myeloma cells. Each protein was confirmed in primary myeloma cells derived from two patients. We have added the western blotting as supplemental data (Figure S1) and added the related description in the main text (Line 99-100, Page 2).

-Line 267. The term unique is not appropriate since no other c-Abl inhibitors have been compared in this study

> Response: As you suggested, we have deleted “unique” in line 271 of the discussion section.

Round 2

Reviewer 2 Report

The revised manuscript adequately addressed the concerns raised by this Reviewer, and it is now acceptable for publication